# Epigenetic Drugs for Cancer and microRNAs: A Focus on Histone Deacetylase Inhibitors

**DOI:** 10.3390/cancers11101530

**Published:** 2019-10-10

**Authors:** Pierre Autin, Christophe Blanquart, Delphine Fradin

**Affiliations:** CRCINA, INSERM, Université d’Angers, Université de Nantes, 44007 Nantes, France; pierre.autin@etu.univ-nantes.fr (P.A.); christophe.blanquart@inserm.fr (C.B.)

**Keywords:** microRNA, HDAC inhibitors, exosome, cancer

## Abstract

Over recent decades, it has become clear that epigenetic abnormalities are involved in the hallmarks of cancer. Histone modifications, such as acetylation, play a crucial role in cancer development and progression, by regulating gene expression, such as for oncogenes or tumor suppressor genes. Therefore, histone deacetylase inhibitors (HDACi) have recently shown efficacy against both hematological and solid cancers. Designed to target histone deacetylases (HDAC), these drugs can modify the expression pattern of numerous genes including those coding for micro-RNAs (miRNA). miRNAs are small non-coding RNAs that regulate gene expression by targeting messenger RNA. Current research has found that miRNAs from a tumor can be investigated in the tumor itself, as well as in patient body fluids. In this review, we summarized current knowledge about HDAC and HDACi in several cancers, and described their impact on miRNA expression. We discuss briefly how circulating miRNAs may be used as biomarkers of HDACi response and used to investigate response to treatment.

## 1. Introduction

In recent decades, non-coding RNAs have been described as key regulators of cellular functions and differentiation. This includes long non-coding RNAs with a size above 200 nucleotides (nt) and small non-coding RNAs (under 200 nt) consisting of numerous subtypes. Micro-RNAs (miRNAs) are endogenous small non-coding RNA of about 19 to 22 nucleotides that modulate gene expression through translational repression, or degradation of the target messenger RNA (mRNA) [1]. A single miRNA has the capacity to inhibit numerous different mRNA targets [2] explaining why miRNAs are potent regulators of gene expression. miRNAs are also important regulators since more than 60% of human genes are regulated by them, as demonstrated by Friedman et al. [3]. In cancer, miRNAs can act as tumor suppressors (TS-miR) or oncogenes (oncomiR), depending on their targets. Recent research has found that miRNAs can not only be detected in tissues but also in all body fluids such as blood, saliva, urine, and milk [4], where they can be used as biomarkers [5]. MicroRNAs harbor attractive features for uses ranging from translation to clinical practices, such as an easy extraction from body fluids, a resistance to molecular degradation by their encapsulation in exosomes, or by their interaction with lipids and proteins, and their easy quantification by different methods including quantitative PCR [6].

In the following sections, we will discuss how miRNAs are regulated by epigenetic drugs, such as histone deacetylase inhibitors (HDACi) used in cancer. We will also succinctly discuss the use of circulating miRNAs as a predictor of response to epigenetic clinical therapies.

## 2. Epigenetic Drugs in Cancer

Epigenetic drugs consist of compounds that inhibit proteins implicated in the writing, the reading, or the erasing of epigenetic marks such as DNA methylation or post-translational modifications (PTM) of histones. Concerning DNA methylation, epigenetic drugs include, for example, the food and drug administration (FDA)-approved decitabine targeting DNMT1 (DNA methyltransferase 1), or AG-221 (or enasidenib), currently tested in a phase III clinical trial (NCT02577406), targeting IDH2 (Isocitrate DeHydrogenase 2), an enzyme providing cofactor for the DNA methylation eraser protein TET1 (ten eleven translocation 1). Concerning PTM, most of the focus has been on histone acetylation erasers that will be described below, but some of them have also been developed against histone methylation writers or erasers, as well as histone acetylation readers, i.e., bromodomain-containing proteins (see review [7]). In this review, we decided to focus on the largest class of epigenetic drugs, the histone deacetylase inhibitors. It is a family of promising epigenetic agents for cancer treatments. Indeed, during cancer initiation, a decrease of histone acetylation leads to the repression of genes resulting in uncontrolled cell proliferation, differentiation and decreased apoptosis. Later, during cancer progression, increasing histone deacetylases (HDAC) activity leads to a loss of cell adhesion, resulting in cell migration, invasion and angiogenesis.

### 2.1. Histone Deacetylase

Previous works have identified 18 deacetylases. These enzymes are classified in four categories depending on homologies with yeast deacetylases, function, localization and substrates (Table 1, more details in the review [8]). Essentially, nucleic HDAC removes the acetyl group on the N-ε-lysine side chain of the histone N-terminal tail, increasing its positive charge, and stabilizing DNA-histone complexes by electrostatic interactions. This induces chromatin compaction and transcription repression. Cytoplasmic HDACs can deacetylate non-histone proteins [9,10,11,12,13,14,15,16,17,18].

### 2.2. Histone Deacetylase Inhibitors

HDACi were first identified from natural sources, currently however, new molecules have been developed with an improved activity and specificity. To date, a high number of compounds are available and evaluated in preclinical or clinical studies. HDACi are classified in four classes according to their chemical structure [19], hydroxamates is the largest one. These compounds are usually pan-HADCi acting in the range of micro to nanomolar concentrations. The well-known members of this family are vorinostat (SAHA), belinostat (PDX101) and panobinostat (LBH589). All of these are approved by the USA food and drug administration (FDA) for the treatment of respectively (i) cutaneous T-cell lymphoma (CTCL) [20], (ii) patients with relapse or refractory peripheral T-cell lymphoma (PTCL) [21], or (iii) multiple myeloma (MM) [22]. Trichostatin A (TSA), the first natural hydroxamate, was excluded from clinical uses due to its high toxicity [23] despite its interesting effects at nanomolar concentrations on cancer cells. The two other groups are benzamides and cyclic peptides which target mainly class I HDAC. The prototypes of these families are entinostat (MS-275) and romidepsin (FK2208) respectively. Romidepsin was approved by FDA for the treatment of CTCL [24] and PTCL [25]. Finally, short chain carboxylic acids, such as valproic acid (VPA) or sodium butyrate (NaBu), inhibit class I and class IIa HDACs.

### 2.3. FDA-Approved Histone Deacetylase Inhibitors

Vorinostat. Vorinostat or suberoylanilide hydroxamic acid (SAHA) is a HDACi belonging to the hydroxamate family, acting on class I and class II HDAC (Table 1 and Table 2). This compound is probably the most used HDACi for preclinical and clinical evaluations. In October 2006, Vorinostat was approved in the USA by the FDA for the treatment of CTCL [26]. When used as a single agent, a poor efficacy was observed on solid tumours [27]. Thus, combination strategies have been or are tested (approximately 134 phase II clinical trials and nine phase III clinical trials in progress in 2019, ClinicalTrials.gov). For examples, Vorinostat is currently evaluated in phase III clinical trials in combination with alkylating agents, proteasome inhibitors, anthracyclines, anti-angiogenics and/or antimetabolites.

Romidepsin. Romidepsin is a bicyclic peptide (Table 2) isolated from a bacteria named *Chromobacterium violaceum* [28,29]. This molecule inhibits mainly class I HDACs (Table 1). Romidepsin is a prodrug that has to be activated in cells to be efficient by the reduction of the disulphide bond included in its structure with the zinc ion present in the HDAC catalytic site. This molecule was approved by the FDA in 2009 for the treatment of patients with CTCL who have received at least one prior systemic therapy [30]. In 2011, the FDA approved romidepsin for the treatment of patients with PTCL who have failed or who were refracted to at least one prior systemic therapy [31]. As for Vorinostat, a poor activity was observed on solid tumours leading to the evaluation of combination strategies in clinic (52 phase II clinical trials and four phase III clinical trials, ClinicalTrials.gov).

Belinostat. Belinostat, a hydroxamate HDACi, presents a broad-spectrum of action (class I and class II HDACi). Belinostat was approved by FDA in 2014 for the treatment of patients with PTCL that was refractory or had relapsed after prior treatment [21,32]. A second phase II clinical trials confirmed these results and showed a better activity of belinostat on PTCL compared to CTCL [33]. The poor activity of belinostat on solid tumor [34] has led to the evaluation of this HDACi in combination with current chemotherapeutic agents (24 phase II clinical trials, ClinicalTrials.gov), notably alkylating agents (cisplatin and carboplatin).

Panobinostat. Panobinostat is a pan-HDACi of the hydroxamate family. A phase III clinical trials, named PANORAMA1, was at the origin of the approval of panobinostat by FDA in 2015, in combination with bortezomib and dexamethasone, for the treatment of patients with multiple myeloma who have received at least two prior regimens, including bortezomib and an immunomodulatory agent [35]. Numerous phase II or III clinical trials, on different cancers, were conducted or are in progress to evaluate the efficacy of this molecule alone or in combination.

## 3. Effect of Histone Deacetylase Inhibitors on Tumor Cells

According to the large number of genes regulated by HDAC, HDACi can affect numerous cellular mechanisms implicated in oncogenic properties of cancer cells. It was notably shown that these molecules induce proliferation arrest, sensitivity to apoptosis, decrease angiogenesis and affect DNA damage repair machinery (Figure 1). Here, we will present only the major pathways affected by HDACi (for more details, see reviews [36,37]).

### 3.1. Cell Cycle

HDACi induced a cell cycle arrest in G0/G1, G1/S or G2/M phase depending on the cancer cell line and on the used HDACi [38]. Induction of expression of the cyclin-dependent kinase (CDK) inhibitor gene *CDKN1A*, coding for p21, seems to be a major mechanism in the cell cycle arrest effect of HDACi even if other CDK inhibitors genes are induced by these molecules [39]. The protein p53 was described as a regulator of p21 expression through binding to its promotor [40]. However, the induction of p21 following HDACi treatment is independent on p53 status of cells [41,42,43] whereas some studies have described an activation of p53 after HDAC inhibition [44,45]. Other mechanisms could explain this observation such as dephosphorylation of retinoblasma protein (Rb) [46,47,48] and inhibition of E2F transcriptional activity [49].

### 3.2. Cell Death

HDACi modulates both the intrinsic and extrinsic pathways of apoptosis. Concerning the extrinsic pathway, HDACi increased Death Receptor (DR4, DR5) expression in cancer cells [49,50,51,52]. Interaction of DR4 and DR5 with tumor necrosis factor (TNF)-super family receptor ligands (Fas-L, TRAIL (TNF-Related Apoptosis Inducing Ligand), TNFα) induced apoptosis by the activation of caspase 8 and 10. Additionally to these regulations, HDACi can also modulate the level of intracellular adaptor molecules, such as the inhibitor of apoptosis named cellular FLICE (Caspase 8)-inhibitory protein (c-FLIP) [50,52,53], or by modulating the interaction between Fas-associated death domain (FADD) and the death-inducing signaling complex (DISC) [50,54]. Intrinsic apoptotic pathways are classically activated by cellular stress stimuli such as free radicals, misfolded proteins or DNA damages. Chemotherapeutic agents can also induce these stress stimuli leading to an increased permeability of the mitochondria and to caspases activation following the release of pro-apoptotic proteins. Intrinsic apoptosis in cells is regulated by the balance of expression of pro-apoptotic (Bak and Bax) and anti-apoptotic BCL-2 proteins (BCL-2, BCL-XL, MCL-1). BH3-only proteins (Bad, Bik, Bid, Bim, Puma, Noxa), a third family of pro-apoptotic proteins, are sensors of cellular stress, and fine tune apoptosis in cells. It is now well established that HDAC inhibition leads to an increasing expression of the pro-apoptotic BCL-2 protein members or BH3-only proteins, such as Bim [55].

### 3.3. Angiogenesis

HDACi have a mainly anti-angiogenic action, modulating angiogenesis by decreasing VEGF expression and hypoxia-inducible factor-1α (HIF1α), but also inducing VEGF (vascular endothelial growth factor) expression in several models of cancers [56,57,58,59,60]. Additional mechanisms were described such as an upregulation of the tumor suppressor gene von Hippel Lindau (VHL) and an alteration of the HSP90 (Heat Shock Protein 90) chaperone function, by modification of its acetylation, all leading to the degradation of HIF1α [61,62]. A direct action of HDACi on the HIF1α stability was described as well, through its acetylation [12]. Finally, HDACi can also affect the capacity of endothelial cells to induce angiogenesis in functional tests [63,64,65,66,67].

### 3.4. DNA Damage

Sensitivity of cancer cells to chemotherapeutic agents, such as alkylating agents or topoisomerase inhibitors, and radiotherapies, can depend on DNA damage repair (DDR) machinery. An increase of the duration of DNA damage induced by irradiation of cancer cells was observed following treatments with HDACi such as VPA, NaBu, vorinostat and MS-275. This demonstrates the incapacity of cells to repair double strand break (DSB) following HDAC inhibition [68,69,70,71]. These observations can be explained by the capacity of HDACi to repress proteins such as Rad50, Ku70 and Ku80, implicated in DDR [71,72]. Others studies showed that TSA, vorinostat and abexinostat can repress *BRCA1* (Breast Cancer 1) and *RAD51* (Recombination Protein A) expressions [73,74] and thereby inhibit the homologous recombination and the non-homologous recombination end joining DDR mechanisms [70,74,75,76]. Finally, cancer cells treatment with HDACi leads to the induction of reactive oxygen species (ROS) which cooperate with the DDR inhibition to induce DNA damages [77,78]. Proposed mechanisms for the induction of ROS by HDACi are a (i) downregulation of the expression of thioredoxin (TRX), reducing protein, (ii) an induction of the expression of the thioredoxin-binding protein-2 (TBP-2) gene as shown in prostate cancer cells [79], and (iii) the induction of the thioredoxin-interacting protein (TXNIP), an inhibitor of TRX, as demonstrated in human gastric cancer cells and HeLa cells [80,81].

## 4. Effect of Histone Deacetylase Inhibitors on microRNA Expressions in Cancer

HDACi treatments can modulate miRNA expressions in tumor cells. Indeed, the first step of miRNA biogenesis is the transcription of the miRNA gene. As classical genes, miRNAs, located outside or inside a coding gene, have their own promoter, TSS (transcription start site), and terminator signals, that are sensitive to epigenetic modifications, such as lysine acetylation which classically opens chromatin structure and enhances transcription activation.

### 4.1. microRNAs Dysregulated in Cancer

miRNA dysregulation in cancer was first reported in 2002, when miR-15 and miR-16 were identified at 13q14.3, a frequently deleted region in chronic lymphocytic leukemia (CLL), leading to the overexpression of their target, i.e., BCL-2 (B cell lymphoma 2) [82]. Different miRNAs have been then labeled as TS-miR (tumor Suppressor miR) or oncomiR based on the nature of their target mRNAs. OncomiRs can repress expression of protein-coding tumor suppressor genes and are frequently upregulated in cancer, whereas TS-miRs target cancer-promoting genes and are downregulated [83].

Let-7c is a one of the most described TS-miRs. It belongs to the let-7 family, highly conserved between species [84]. Let-7c is frequently downregulated in cancer, or even deleted since it is located in a region of frequent homozygous deletion [82]. Let-7c targets various oncogenes and cancer related genes such as IL6-R (interleukin-6 receptor) [85] or E2F5 (E2F transcription factor 5) [86] (Table 3). Its downregulation is also associated with poor prognosis in non-small cell lung carcinoma (NSCLC) [87], in colorectal cancer, or in metastatic prostate cancer [88].

The miR-17-92 cluster highly conserved among species, comprises six miRNAs (miR-17-5p, miR-18a, miR-19a, miR-20a, miR19b-1 and miR-92a-1), that are overexpressed in many human cancers. miR-18a is one of the most expressed of this cluster, and is considered as an oncomiR. It has been found to be upregulated in breast cancer, head and neck squamous cell carcinoma, esophageal squamous cell carcinoma, gastric carcinoma, pancreatic carcinoma, hepatocellular, and colorectal carcinoma [89]. Interestingly the concentration of miR-18a in plasma or serum of patients with cancer is much higher than that of healthy persons [89]. So aberrant expression of miRNA might serve as a biomarker of cancer or to evaluate cancer response to treatment in non-invasive liquid biopsy.

Over these two examples, numerous miRNAs are dysregulated in malignancies and many data are currently available on their expression for diagnostic or prognostic uses (see review [90]).

### 4.2. miRNA Regulated by Histone Deacetylase Inhibitors

A few years back, there were discrepancies about miRNA expression modifications by HDACi in tumors [100,101] probably resulting from different parameters such as cell lines and/or concentrations used. Since then, involvement of miRNAs in HDACi effects on tumors have been well documented. In vitro, miRNAs have an important role in HDACi effects such as in colorectal cancer, where it has been demonstrated that vorinostat modulates not less than 275 miRNAs resulting in a myriad of possible targets and pathways affected [102]. Moreover, in some cases, miRNAs modulated by HDACi have been correlated with tumor stage or clinical outcome. In this section, we will describe the main miRNAs involved in HDACi effects on tumors. In the current state of the art, it is challenging to find a link between miRNAs, HDACi and/or a specific cancer or pathways, and so the following section is ordered regarding both HDACi approval and actions of miRNAs involved (namely, TS-miRs and oncomiRs).

#### 4.2.1. FDA-Approved HDACi and miRNAs

As mentioned earlier, only four HDACi have been approved by the FDA, namely: Vorinostat (SAHA), Panobinostat (LBH589), Belinostat (PXD101), and Romidepsin (FK288). Several studies on various tumor models have been led to understand the mechanisms of these molecules and especially, the importance of miRNAs in tumor-suppressive effects of these HDACi (Table 4).

HDACi-induced TS-miRs. Treatment by HDACi leads to an increase expression of TS-miR from the let-7 family in several models. Similarly, in hepatocellular cancer, in vitro and in vivo treatment with vorinostat or panobinostat triggers let-7b upregulation, leading to the downregulation of BCL-XL, TRAIL (tumor necrosis factor (ligand) superfamily, member 10) or the oncoprotein HMG2A (high mobility group box 3) [103] (Figure 2). Other upregulation, induced by vorinostat, of almost all let-7 family members have been reported in ovarian cancer by Balch et al. [104] as well as in renal cancer by Pili et al. [105]. Conversely, studies described downregulation induced by vorinostat of let-7 family members (let-7b, let-7c, let-7f) in other types of tumors such as lung [106] and thyroid cancers [107]. However, these last studies only described miRNAs modification without going further into let-7-related mechanisms and functions. These discrepancies can also be a consequence of the methods used to purify and screen miRNAs.

Another example of TS-miR is the miR-200 family, consisting of five members divided into two clusters, namely, miR-200b, -200a, and -429 (cluster I); and miR-200c and -141 (cluster II). They are often found to be lost in cancers with different pathways involved [108]. This family appears to be linked with HDACi effects especially in breast cancers were two studies described upregulation of miR-200a and miR-200c induced by vorinostat resulting in (i) an upregulation of antioxidant pathway Nrf2 and (ii) a decrease of proliferation, invasion and migration in tumor cell lines [109,110].

Other miRNAs have been described as playing a crucial role in these HDACi-induced modifications depending on cancer type. Panobinostat treatment has lead to increased cell senescence through miR-31 in breast cancer cells [111]. In pancreatic cancer cells, vorinostat induced many modifications of cell phenotype through miR-34a [112]. One of the most common mechanisms described in the literature is the ability by several HDACi to increase apoptosis in various tumor cell lines in a miRNA-dependent manner (leukemia, lymphoma, pancreatic cancer). This mechanism has been explained by a HDACi-induced upregulation of several miRNAs such as miR-15a, miR-16, miR-34a or miR-195 leading to a downregulation of their target mRNAs mainly (but not only) from the BCL-2 family in vitro and in vivo in mice [104,113,114] (Figure 2).

HDACi-induced oncomiR. The role of miR-17~92 cluster members in promoting tumorigenesis has been widely demonstrated and thus, effects of HDACi on these miRNAs has also been evaluated. Even though the oncogenic role of the miR-17~92 cluster has been largely described, the six miRNAs composing this cluster are not equivalent when it comes to promoting tumorigenesis. Consistently, HDACi affect these miRNAs towards repressing the tumor-promoting tendency of this cluster. In the literature, vorinostat mechanisms often seem to rely on miR-17~92 miRNAs. Indeed, in lymphoma, vorinostat decreases miR-17-5p and miR-18 through c-myc, leading to more sensitivity to apoptosis of tumor cells [115] (Figure 2). In another hematopoietic cancer, Lepore et al. demonstrated that vorinostat in human leukemia cell lines, leads to increasing apoptosis through repression of BARD-1 (BRCA1 associated RING domain 1) protein. This vorinostat-induced BARD-1 repression was due to the modulation of several miRNAs within the cell including especially, and surprisingly, an upregulation of miR-19a and miR-19b [116]. This highlights the fact that despite its role as an onco-miR in some cases, treatment mechanisms involving miR-19 are diverse and still need to be fully elucidated. Modulation of this cluster by HDACi has also been confirmed in solid tumors [117]. miR-20a and other miRNAs from the miR-17~92 cluster showed an altered expression in hepatocellular cancer, by vorinostat resulting in an upregulation of MICA protein levels and a better recognition of these tumor cells by innate immune cells and especially NK cells [118]. Moreover in colorectal and renal cancer, the decreased cell proliferation induced by vorinostat have been linked to a repression of the miR-17~92 cluster expression [119,120].

#### 4.2.2. Other HDACi and miRNAs

There are plenty of HDACi that have not yet been approved by the FDA. Some of them are involved in phase III clinical trials such as Valproic acid to treat cervical and ovarian cancers or Tacedinaline for multiple myeloma and lung cancers [124]. Others are in earlier stages but nonetheless, interesting studies have been done to strengthen the close relationship between HDACi effects and miRNA-related mechanisms (Table 5).

Firstly, and expectedly, some non-approved HDACi act on similar pathways and miRNA clusters that the ones authorized by the FDA. In lung cancer, the let-7 family miRNAs are also upregulated by TSA, leading to increased cell cycle arrest and apoptosis of tumor cells compared to adjacent non-tumorous lung tissue [125]. Similarly, in lymphoma, let-7a alongside with other miRNAs are upregulated by Romidepsin, decreasing anti-apoptotic proteins such as BCL-2 (B-cell CLL/lymphoma 2) and BCL-XL (BCL2-like 1) [126]. The miR-17~92 cluster members have been described to be regulated by butyrate in colorectal cancer [127] and the miR-200 family is involved in reducing tumor cell proliferation in NSCLC and SCLC (Small Cell Cancer of the Lung) treated with Entinostat (MS275) [128]. miR-34 and miR-15a, described below, are also upregulated by AR42 in ovarian cancer, which trigger a cascade of pathways leading to a decrease of Wnt receptor signaling and EMT (epithelial mesenchymal transition), and an increase of negative regulation processes of cell cycle and apoptosis [104]. The same HDACi in pancreatic cancer, decreases p53 and cyclins protein levels thanks to variation of miR-30d, miR-33, and miR-125b, leading to the inhibition of cell proliferation, invasion and tumor growth, and to an increase of ROS generation, DNA damage and apoptosis [129]. Mocetinostat, a clinical phase II HDACi, has been described as involving miR-31 in the inhibition of E2F6 (E2F transcription factor 6), leading to apoptosis of the prostate tumor cells [130]. Another HDACi, OBP-801, has been described as inducing, both in vitro and in vivo (mice), a decrease of tumor cell growth involving an upregulation of miR-320a [131]. In the same study, they also identified that miR-320a was almost not modified by other HDACi such as SAHA or TSA, highlighting the mechanistic specificities of HDACi.

#### 4.2.3. Clinical Relevance

Interestingly, miRNAs modulated by HDACi have been proven to have importance for clinical outcomes, such as miR-200c and miR-203 in pancreatic adenocarcinomas directly resected from patients. Indeed, the group with no recurrence within six months exerted a much higher level of these two miRNAs than the “recurrence group” [146]. In primary resected tumors, it has been demonstrated that miR-200c and miR-203 may also have a biomarker relevance. Indeed, Hibino et al. showed a significant association between non-recurrence and a high expression of miR-203 and miR-200c [114]. In a clinical study on renal cancer patients, miR-605 was directly targeted by a combination of vorinostat and bevacizumab, an antibody targeting growth factors. They demonstrated that this miRNA was upregulated in treatment responders at baseline and that it was downregulated after treatment ([105]; clinical trial NCT00324870). This is explained by the fact that these miRNAs, modified by epigenetic drugs such as HDACi, crosstalk with other proteins such as p53 for instance and are, therefore, able to shift pathways into anti-tumor outcomes for the cell. To further confirm the importance of miRNAs and their relevance in clinics, several ongoing trials plan to investigate miRNA involvement in HDACi-related effects such as Belinostat in carcinoma patients (NCT00926640), or vorinostat in bladder and renal cancers (NCT00926640).

Overall, as previously noticed, it appears difficult to bring out common mechanisms whether it is regarding HDACi molecules, tumor types or miRNAs involved. Conventional clusters such as let-7, miR-17~92, miR-200 are often described to be modified by HDACi but other less studied miRNAs have also been recently described. As expected, effects described are consistent with pathway modifications described in Section 3 of this review. Finally, most of the aforementioned articles have functionally tested miRNAs (mostly with miRNA mimic and/or anti-miR), describing both the importance and the need of these miRNAs to be involved in HDACi-induced mechanisms.

### 4.3. Histone Deacetylase Inhibitors and Circulating miRNAs

As mentioned earlier, miRNAs modulated by HDACi can also be screened in body fluids even if few studies have investigated this characteristic. As a proof of concept, Pili et al. evaluated the modulation of circulating miRNAs in clear-cell renal cell carcinoma (ccRCC) patients under a combinatory treatment of vorinostat and bevacizumab (a humanised monoclonal antibody that neutralises VEGF) [105]. They observed in responder patients an upregulation of miR-20a and miR-let-7b and a downregulation of miR-142-3p, miR-154, miR-605 and miR-199a-5p after treatment. Conversely, miR-605 was upregulated after treatment in progressor patients. Interestingly, this miRNA participates in the p53 network [147] and is frequently upregulated or mutated in cancers [148,149]. To our knowledge, this is the only study about the use of circulating miRNAs as a prognostic biomarker of HDACi response, even if it is a conventional approach for other anti-tumor treatments, as recently described in the review of Najminejad et al. [150] and Pardini et al. [151]. We believe however, that it can be a promising approach since miRNAs are stable and easily detectable in all body fluids. Indeed, circulating miRNA are protected from RNase activity through their conjugation with proteins, their inclusion in lipid or lipoprotein complexes or through their loading in exosomes/microvesicles. Exosomes are small intraluminal vesicles that are 50–150 nm in diameter. They are generated inside multivesicular endosomes (MVB) [152] that fuse with cell membranes and release the vesicles into the extracellular space. Exosomal miRNAs participate in intercellular communication (Figure 3). Uptake by normal cells of the exosome cargo secreted by cancer cells can affect the behavior of recipient cells in various ways that provide benefits to the tumor. Several studies have described how these exosomal miRNAs, induced or not by treatment, participate in tumor immune escape [153,154] or drug resistance [155,156].

## 5. Conclusions

Many miRNAs have shown different expression levels in response to HDACi treatment. Some of them can potentiate the anti-tumor response, or on the contrary, decrease it. Since tumor cells release miRNAs through exosomes that can be detected in all body fluids, such as plasma, urine or saliva, analysis of circulating miRNAs in patient liquid biopsies provides promising biomarkers to monitor drugs in patients. However, to date, it is still challenging to accurately identify clinically relevant miRNAs due to the lack of standardization in their extraction or in the conservation of biopsy, which greatly affects the stability of miRNAs.

## Figures and Tables

**Figure 1 cancers-11-01530-f001:**
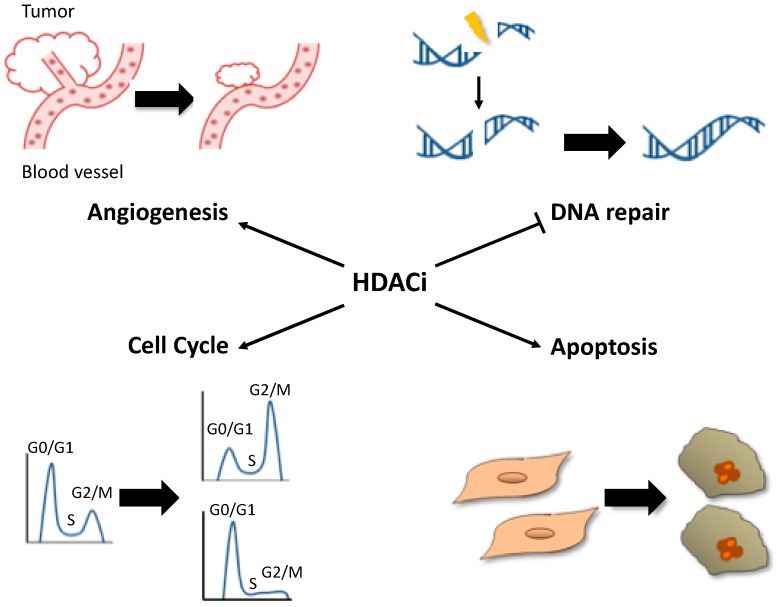
The main cellular processes affected in cancer cells by HDACi treatments. The decrease of histone acetylation by HDACi leads to the modification of the expression of several genes implicated in oncogenic properties of cancer cells. From top left to bottom right, HDACi reduces angiogenesis and tumor growth, HDACi improves treatments by inhibiting DNA repair, HDACi induces cell cycle arrest and stimulates apoptosis.

**Figure 2 cancers-11-01530-f002:**
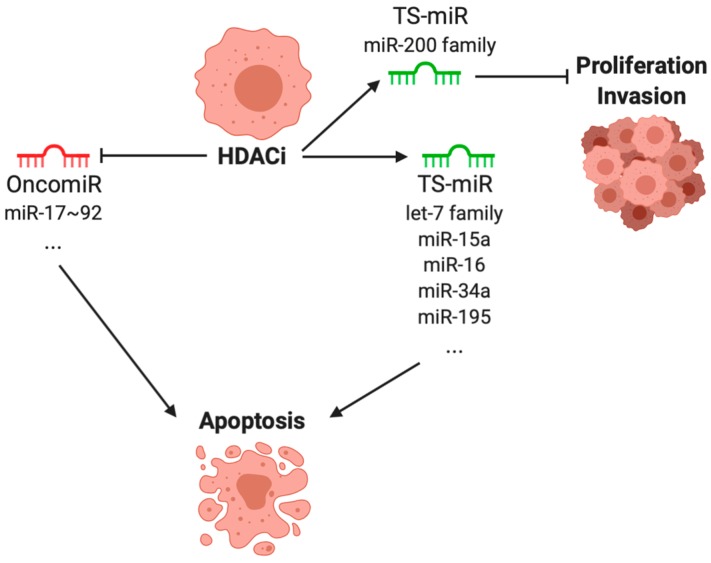
miRNAs modulated by HDACi treatments in cancer. HDACi upregulate TS-miR and downregulate oncomiR to inhibit proliferation and metastasis and to favor apoptosis.

**Figure 3 cancers-11-01530-f003:**
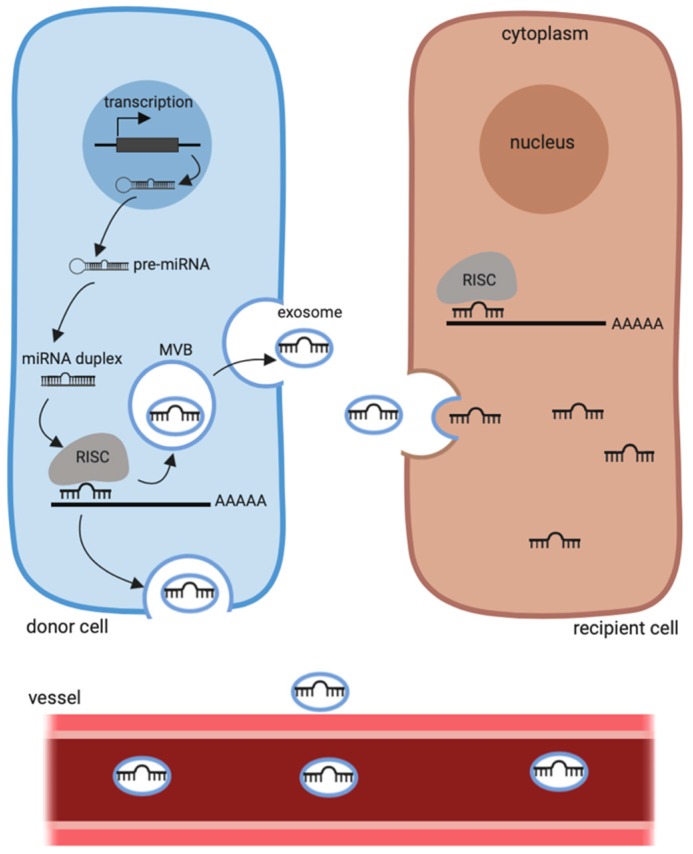
miRNA biogenesis pathway. miRNA is transcribed in the nucleus and then cleaved numerous times to conduct to a mature single strand miRNA included in the RISC complex. miRNA may regulate gene expression in the cell but also in other cells by their encapsulation in microvesicles such as exosomes. miRNA may also be disseminated through the bloodstream. MVB: endosomal MultiVesicular bodies, RISC: RNA-induced silencing complex.

**Table 1 cancers-11-01530-t001:** Classification of histone deacetylase inhibitors.

Class	Targeted Histone Deacetylases (HDACs)	Localization	Zn^2+^	Expression
I	1, 2, 3, 8	Nucleus	Yes	Ubiquitous
IIa	4, 5, 7, 9	Nucleus and cytoplasm	Yes	Tissue specific
IIb	6, 10	Cytoplasm	Yes	Tissue specific
III	Sirtuins 1–7	Nucleus, cytoplasm and mitochondria	No	Variable
IV	11	Nucleus and cytoplasm	Yes	Ubiquitous

**Table 2 cancers-11-01530-t002:** Structure and applications of the four food and drug administration (FDA)-approved histone deacetylase inhibitors.

Name	Structure	Year of Approval	Application
Vorinostat	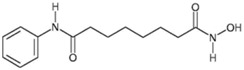	2006	Cutaneous T Cell Lymphoma
Romidepsin	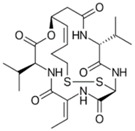	2009 2011	Cutaneous T Cell Lymphoma Peripheral T Cell Lymphoma
Belinostat	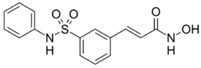	2014	Peripheral T Cell Lymphoma
Panobinostat	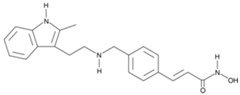	2015	Multiple Myeloma

**Table 3 cancers-11-01530-t003:** Let-7c target genes described in cancer diseases.

Disease	Targets	Function	Reference
Glioma	E2F5	Control of cell cycle	[86]
Melanoma	CALU	Protein folding and sorting	[91]
Lung cancer	RAS	Oncogene	[92]
NSCLC	ITGB3/MAP4K3	Metastatic abilities	[87]
Cholangiocarcinoma (CCA)	IL6-R	Immune response	[85]
	EZH2/DVL3/βcatenin	Metastatic abilities	[93]
Oral squamous cell carcinoma	IL8	Immune response	[93]
Lung adenocarcinoma	BCL-XL	Inhibitor of cell death	[94]
Ovarian carcinoma	CDC25A	Control of cell cycle	[95]
Hepatocellular carcinoma (HCC)	CDC25A	Control of cell cycle	[96]
Colorectal cancer	MMP11/PBX3	Metastatic abilities	[97]
Erythroleukemia	PBX2	Transcription	[98]
Breast cancer	ERCC6	Transcription/excision repair	[99]

E2F5: E2F transcription factor 5, CALU: calumenin, ITGB3: integrin beta 3, MAP4K3: mitogen-activated protein kinase kinase kinase kinase 3, IL6-R: Interleukin 6 receptor, EZH2: enhancer of zeste 2 polycomb repressive complex 2 subunit, DVL3: dishevelled segment polarity protein 3, IL8: Interleukin 8, BCL-XL: BCL2-like 1, CDC25A: cell division cycle 25A, MMP11: matrix metallopeptidase 11, PBX3: pre-B-cell leukemia homeobox 3, PBX2: pre-B-cell leukemia homeobox 2, ERCC6: excision repair cross-complementation group 6

**Table 4 cancers-11-01530-t004:** microRNAs regulated by the four FDA-approved histone deacetylase inhibitors in cancers.

Cancers	HDACi	miRNAs	miRNA Targets	Pathways	Ref.
Breast	Vorinostat	 miR-200a	 Keap1	 Nrf2 antioxidant pathway	[109]
 miR-200C	 CRKL	 Invasion	[110]
		 Migration
Panobinostat	 miR-31, miR-125a, miR-125b, miR-205, miR-141, miR-200c	 NF-kB inducing kinase, ITGA5, SEPHS1, RSBN1, TFDP1	 Cellular senescence	[111]
 BMI1 and EZH2 (indirect)
Colorectal	Vorinostat	Changes in 275 out of the 723 studied human miRNAs	see article for predicted targets	[102]
 miR-17-92 cluster	 PTEN	Proliferation (opposite effects depending on members of the cluster)	[119]
	 mRNA levels of c-MYC, E2F1, E2F2 and E2F3
HCC	VorinostatPanobinostat	 let-7b	 p21	 E2F1 transcriptional activity	[103]
 MYC, MET, HMGA2, TRAIL, BCLX	 Cell proliferation
Vorinostat	 miR-17, miR-18a, miR-19a, miR-20a, miR-93, miR-106b	 MICA, MICB	 Recognition of tumor by innate immune cells	[118]
Leukemia	VorinostatRomidepsin	 miR-15a, miR16, miR29b	 MCL1, BCL-2	 Apoptosis	[113]
Vorinostat	 23 miR (e.g. miR-19a, miR-19b)	 BARD-1	 Sensitivity to vorinostat	[116]
 26 miR (see article)		 Apoptosis
 miR-196a	 BCR/ABL	 Transcriptional activity of the pluripotency factors	[121]
 Cell cycle progression genes
 Sentivity to imatinib mesylate (a Tyrosine Kinase inhibitor)
Panobinostat	miR-26a, miR-133a, miR-181b, miR-182, miR-200c, miR-211, miR-320a, miR-320c, miR-423-5p, miR-638, miR-877, miR-1307, miR-1308, miR-1268miR-516a-3p, miR-605	 Homologous recombination repair pathway (RAD51, BRCA1, NBS1)	 Homologous recombination repairdelay DNA repair  Sensitivity to CNDAC (prodrug used in AML)	[122]
Lung	Vorinostat	 let7b, miR-17*, miR-92a	expected targets for each miR listed in the article	[106]
 miR-373	 LAMP1, VSP4B, IRAK2, BRMS1L, SYDE1, CYBRD1, PDIK1L, C10orf46, TGFBR2	Associated with poorer disease-free survival	[123]
Lymphoma	Vorinostat	 miR-15b, miR-17-3p, miR-17-5p, miR-18, miR-34a, miR-155	 c-myc	 Sensitivity to apoptosis	[115]
Ovarian	Vorinostat	 Let-7, miR-99, miR-100, miR-125… (see figure in article)	[104]
Pancreatic	Vorinostat	 miR-34a	 Cyclin D1, CDK6, SIRT1, survivin, BCL-2, VEGF, Notch pathway	 Cell proliferation, stem cell renewal, invasivness	[112]
 p21/CIP1, acetylated p53, PUMA	 Apoptosis, cell cycle arrest

CRKL: v-crk avian sarcoma virus CT10 oncogene homolog-like, NF-kB: nuclear factor of kappa light polypeptide gene enhancer in B-cells 1, ITGA5: integrin, alpha 5, SEPHS1: selenophosphate synthetase 1, RSBN1: round spermatid basic protein 1, TFDP1: transcription factor Dp-1, BMI1: BMI1 proto-oncogene, polycomb ring finger, EZH2: enhancer of zeste 2 polycomb repressive complex 2 subunit, PTEN: phosphatase and tensin homolog, E2F: E2F transcription factor, p21/CIP1: cyclin-dependent kinase inhibitor 1A, MET: MET proto-oncogene, receptor tyrosine kinase, HMGA2: high mobility group AT-hook 2, TRAIL: tumor necrosis factor (ligand) superfamily, member 10, BCLX: BCL2-like 1, MICA/B: MHC class I polypeptide-related sequence A/B, MCL1: myeloid cell leukemia 1, BCL-2: B-cell CLL/lymphoma 2, BARD-1: BRCA1 associated RING domain 1, BCR: breakpoint cluster region, ABL: ABL proto-oncogene 1, non-receptor tyrosine kinase, RAD51: RAD51 recombinase, BRCA1: breast cancer 1, early onset, NBS1: nibrin, LAMP1: lysosomal-associated membrane protein 1, IRAK2: interleukin-1 receptor-associated kinase 2, BRMS1L: breast cancer metastasis-suppressor 1-like, SYDE1: synapse defective 1, Rho GTPase, homolog 1, CYBRD1: cytochrome b reductase 1, PDIK1L: PDLIM1 interacting kinase 1 like, TGFBR2: transforming growth factor, beta receptor II, CDK6: cyclin-dependent kinase 6, SIRT1: sirtuin 1, VEGF: vascular endothelial growth factor A, PUMA: BCL2 binding component 3. Arrows indicate decrease (

) or increase (

) of either miRNA or target and their associated pathway.

**Table 5 cancers-11-01530-t005:** microRNAs modulated by histone deacetylase inhibitors used in cancer models.

Cancers	HDACi	miRNAs	miRNA Targets	Pathways	Ref.
Breast	LAQ824	 miR27a (≈40% of miRNAs modulated)	 RYBP/DEDAF, ZBTB10/RINZF		[101]
TSA	 22 miR among which: miR-1, miR-143, miR-144, miR-191-3p, miR-202-5p…	(predicted targets for each miRNAs provided in the article)	[132]
	 10 miR among which: miR-500, miR-645, miR-512-3p, miR-613…		
	(see article for complete listing)		
TSA, VPA NaBu…	 miR125-a	 HDAC5 mRNA	 apoptosis	[133]
CCA	CG200746	 miR-22-3p, miR-22-5p, miR-194-3p, miR-194-5p, miR-210-3p, miR-509-3p	expression induced in treated cells	 tumor growth  proliferation	[134]
Colorectal	PBA	 miR-9, miR-127, miR-129, miR-137			[135]
Butyrate	 18 miRNAs	 p21 protein expression	 proliferation	[127]
 26 miRNAs (among which miR-17-92a, miR-18b-106 and miR25-106b clusters)
Entinostat (MS-275)	 pri and mature miR-21			[136]
Gastric carcinoma	TSA	 miR-375	 PDK1, XIAP, 14-3-3ζ (YWHAZ), cIAP-2 (BIRC3)	 Tumor cell viability	[137]
BCL2L11 (Bim)	 apoptosis
HCC	TSA	 miR-449	 c-MET	 cell proliferation  apoptosis	[138]
Sodium valproate	 miR-889	 MICB	 sensitivity to NK cell-mediated lysis	[139]
Leukemia	valpromide(=VPA analog)	 miR-144, miR-451, miR-155 (all cells)	 GATA-1	 erythropoiesis impairment	[140]
	 GATA-2
 miR-15a, miR-16, miR-222 (some cells)	 ETS family (PU.1, ETS-1, GABP-α, Fli-1)	 megakaryocyte features
Lung	Entinostat (MS275)	 miR-200a	 KEAP1/NRF2	 cell growth	[128]
TSA	 Let-7, miR-15a, miR-16-1		  proliferation and apoptosis	[125]
				induce cell cycle arrest
Lymphoma	RGFP966	 miR-15a, miR-195, let-7a (in vitro and in vivo)	 BCL-2, BCL-XL	 apoptosis	[126]
Melanoma	4PBA (or 5Aza, 5Aza + 4PBS)	 miR-34b, miR-132, miR-142-3p, miR-200a, miR-375, miR-489		 Proliferation, invasion	[141]
	 wound healing
		changes in cell morphology
Multiple Myeloma	AR-42	 miR-9-5p	 CD44		[142]
Ovarian	AR42	 miR-15a, miR-34, …(see figure in article)	 WT1, PAX2, GATA6, APO2/TRAIL…(see article)	 EMT, Canonical Wnt R signaling  Negative regulation of cell cycle processes, extrinsic apoptosis	[104]
Pancreatic	AR-42	 miR-30d, miR-33, miR-125b	 p53, cyclin B2, CDC25B	 Invasion, tumor growth	[129]
Prostate	Mocetinostat	 miR-31	 E2F6	 apoptosis	[130]
OBP-801	 miR-320a in vitro and in vivo (rat)	 PSA, androgen receptor	 Viability, cell growth, cell proliferation, prostate tumorigenesis (in vivo)	[131]
Tongue	TSA (or Doxorubicin, 5-fluorouracil, etoposide treatments)	 miR-375	 CIP2A, MYC, 14-3-3z, E6AP, E6, E7	 cell proliferation, migration and invasion	[143]
	 p21, p53, RB
Various models	PBA (and 5-Aza-CdR)	 17 miR/313 studied (see article for details)	 BCL6 (suggested)		[144]
TSA	 miR132/212	 MeCP2		[145]

NaBu, Sodium Butyrate, E2F: E2F transcription factor, p21/CIP1: cyclin-dependent kinase inhibitor 1A, MET: MET proto-oncogene, receptor tyrosine kinase, BCL-2: B-cell CLL/lymphoma 2, RYBP/DEDAF: RING1 and YY1 binding protein, ZBTB10/RINZF: zinc finger and BTB domain containing 10, HDAC5: histone deacetylase 5, PDK1: pyruvate dehydrogenase kinase, isozyme 1, XIAP: X-linked inhibitor of apoptosis, 14-3-3ζ (YWHAZ): tyrosine 3-monooxygenase/tryptophan 5-monooxygenase activation protein, zeta, cIAP-2 (BIRC3): baculoviral IAP repeat containing 3, BCL2L11 (Bim): BCL2-like 11, MICB: MHC class I polypeptide-related sequence B, GATA: globin transcription factor, KEAP1/NRF2: kelch-like ECH-associated protein 1, BCL-XL: BCL2-like 1, WT1: Wilms tumor 1, PAX2: paired box 2, APO2/TRAIL: tumor necrosis factor receptor superfamily, member 10a, PU.1: Spi-1 proto-oncogene, ETS-1: v-ets avian erythroblastosis virus E26 oncogene homolog 1, GABP-α: GA binding protein transcription factor, alpha subunit 60kDa, CDC25B: cell division cycle 25B, PSA: prostate specific antigen, CIP2A: cancerous inhibitor of PP2A, RB: retinoblastoma 1, BCL6: B-cell CLL/lymphoma 6, MeCP2: methyl CpG binding protein 2. Arrows indicate decrease (

) or increase (

) of either miRNA or target and their associated pathway.

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
