# Peer review of "Epigenetic Drugs for Cancer and microRNAs: A Focus on Histone Deacetylase Inhibitors"

_cancers, 2019, doi:10.3390/cancers11101530_

Round 1

Reviewer 1 Report

The review was well-rewritten and organized. Two major topics which will strengthen this review is the tumor immunology and drug resistance (or cancer stem cells).  

Author Response

We thank the reviewer for its so positive comment.

Resistance to HDACi is often seen and the basis of this resistance remains largely unknown. To our knowledge no miRNAs have been implicated in this process until now. This is however an interesting point which deserves to be studied.

Tumor immunology is also an interesting topic. Only one article has investigated this point (Yang et al 2015, ref 118 in our manuscript). They showed that vorinostat upregulated the transcription of MICA/B by the downregulation of the miR-17-92 cluster. MICA/B are ligands of the natural killer (NK) cell activating receptor NKG2D, thus increasing the susceptibility of tumor cell to lysis by NK cells.

Results of this article are discussed in section 4.2.1.

Reviewer 2 Report

HDAC inhibitors are clinically approved epigenome modifying small molecules that are effective in treating haematogical cancers such as CTCL. These inhibitors target HDACs in the cells, which in turn leads to transcriptional regulation of protein-coding genes that drive cell cycle arrest, cell death and differentiation in tumour cells. While miRNAs have been identified as key regulators of tumour suppressors and oncogenes, it is relatively unknown which miRNAs are regulated by HDACs and whether the expression of any miRNAs correlate to HDAC inhibitor sensitivities. This literature review summarized some of the latest findings in the field to address the potential of miRNAs as biomarker for HDAC inhibitor sensitivities.

This topic of the review is of significant interests to the field of HDACs and HDAC inhibitors, however, the manuscript requires a considerable efforts in language editing. The manuscript also requires specific attention towards the overall and section structures. It is recommended that authors consider the following structure that would help the review more readable to the readers.

HDACs, HDAC inhibitors and cellular phenotypes HDACs HDAC inhibitors for both approved and ones in development Mode of actions for their anti-tumour activities (and describe key protein coding genes involved in each processes) Cell death Cell cycle arrest Other phenotypes: cell differentiation, angiogenesis and DNA damage miRNAs that are dysregulated in cancers miRNAs that are regulated by HDACs and HDAC inhibitors Potential miRNA biomarkers

I recommend that the authors spend more time editing and rewriting the papers with the existing content. The table are good but perhaps include a few figures. Some sections can be removed as the content is too general. I am more than happy to read the paper again to provide more specific comments. It is rather difficult to make specific comments on the current form of the manuscripts as there are too many to cover.  

Author Response

We thank the reviewer for its suggestions which greatly improve our manuscript. As requested by the reviewer, we have removed general sections about miRNAs and HDACs. We have also followed the proposed structure and added a figure.

Reviewer 3 Report

Epigenetic drugs for cancer and microRNAs

General comments:

The review seeks to summarize the effect of HDACi on microRNA expression and the impact of these changes on tumour biology and as biomarkers of HDACi response.  While the changes in microRNA expression induced by HDACi in different tumour cell lines are well documented, the review falls short in cohesively helping the reader interpret which are the key changes required for the anti-tumour effects of these drugs.

For instance, on line 430 the Authors acknowledge that it is difficult to bring out common mechanisms in regards to HDACi-induced microRNA changes.  While I acknowledge this is challenging, this is ultimately the purpose of the review.  Otherwise, as currently written, it merely acts to highlight the vast array of microRNA changes induced by the drugs, and the reviewer is left wondering which effects on microRNA biology induced by HDACi, if any, are particularly relevant.  In my opinion the Authors need to do a much better job in trying to make sense of the multiple studies out there in order to make this a review of value to the readers. 

Review structure

A significant part the review covers rudimentary knowledge that should be well known to most cancer biologists.  This includes a summary of microRNA biology and the role of microRNAs in cancer, well understood concepts such as gene regulation by DNA methylation (2.3) and histone modification.  Details regarding the classes of HDACs (3.1), HDACi (3.2, 3.3) have been reviewed several times previously, as have the effects of HDACi on tumour cells (3.4). As such, sections 1-3 provide little new information above what is well known or has been reviewed extensively already.

Section 4 of the review (page 8, line 298) is when the review finally begins to address what is suggested in the title and abstract.

Specific points:

Several examples of microRNAs altered in expression following HDACi treatment are cited. However, the authors don’t delve into great detail as to which studies have shown these changes are directly important for HDACi-induced phenotypic changes.  It would be good to know which studies use microRNA knockdown or rescue experiments to determine direct cause and effect, and which are correlative? 

Abstract states that the review will describe how circulating miRNAs can be used as biomarkers of HDACi response.  However this is ultimately limited to a single study (Ref 18).  What about CTCL where these drugs are approved?     

Title refers to epigenetic drugs, but the review is exclusively focussed on HDACi.  Consider rewording.

The title is fairly uninformative.

The Tables are very difficult to digest.

Author Response

Reviewer 3

Epigenetic drugs for cancer and microRNAs

General comments:

The review seeks to summarize the effect of HDACi on microRNA expression and the impact of these changes on tumour biology and as biomarkers of HDACi response.  While the changes in microRNA expression induced by HDACi in different tumour cell lines are well documented, the review falls short in cohesively helping the reader interpret which are the key changes required for the anti-tumour effects of these drugs.

For instance, on line 430 the Authors acknowledge that it is difficult to bring out common mechanisms in regards to HDACi-induced microRNA changes.  While I acknowledge this is challenging, this is ultimately the purpose of the review.  Otherwise, as currently written, it merely acts to highlight the vast array of microRNA changes induced by the drugs, and the reviewer is left wondering which effects on microRNA biology induced by HDACi, if any, are particularly relevant.  In my opinion the Authors need to do a much better job in trying to make sense of the multiple studies out there in order to make this a review of value to the readers. 

The two main pathways regulated by HDACi-induced miRNAs are the apoptosis and proliferation/invasion pathways. However, miRNAs differ between cancer type, as well as their target, but ultimately, they regulate either genes implicated in apoptosis or in proliferation. We have synthetized this conclusion in Figure 2.

Review structure

A significant part the review covers rudimentary knowledge that should be well known to most cancer biologists.  This includes a summary of microRNA biology and the role of microRNAs in cancer, well understood concepts such as gene regulation by DNA methylation (2.3) and histone modification.  Details regarding the classes of HDACs (3.1), HDACi (3.2, 3.3) have been reviewed several times previously, as have the effects of HDACi on tumour cells (3.4). As such, sections 1-3 provide little new information above what is well known or has been reviewed extensively already.

Section 4 of the review (page 8, line 298) is when the review finally begins to address what is suggested in the title and abstract.

The reviewer is right. We have modified the structure of our manuscript and removed sections with a too general content.

Specific points:

Several examples of microRNAs altered in expression following HDACi treatment are cited. However, the authors don’t delve into great detail as to which studies have shown these changes are directly important for HDACi-induced phenotypic changes.  It would be good to know which studies use microRNA knockdown or rescue experiments to determine direct cause and effect, and which are correlative? 

For most of the studies (12/16 in FDA-approved and 15/23 in non-approved parts), functional experiments have been conducted. For example, Xie et al. (2018, ref 140) realized overexpression and knockdown experiments to determine the role of the identified miRNAs. We have clarified this point in the section 4.2.

Abstract states that the review will describe how circulating miRNAs can be used as biomarkers of HDACi response.  However this is ultimately limited to a single study (Ref 18).  What about CTCL where these drugs are approved?     

This is an interesting point. To our knowledge, there is no study about circulating miRNAs induced in response to HDACi in CTCL.

Title refers to epigenetic drugs, but the review is exclusively focussed on HDACi.  Consider rewording.

The title is fairly uninformative.

Accordingly, we have completed our title.

Round 2

Reviewer 3 Report

The revised version of the review is an improvement on the previous version.